# Conformal Embedding Flows: Tractable Density Estimation on Learned Manifolds

**Brendan Leigh Ross** [1]   **Jesse C. Cresswell** [1]

## Abstract

Normalizing flows are generative models that provide tractable density estimation by transforming a simple distribution into a complex one. However, flows cannot directly model data supported on an unknown low-dimensional manifold. We propose Conformal Embedding Flows, which learn low-dimensional manifolds with tractable densities. We argue that composing a standard flow with a trainable conformal embedding is the most natural way to model manifold-supported data. To this end, we present a series of conformal building blocks and demonstrate experimentally that flows can model manifold-supported distributions without sacrificing tractable likelihoods.

## 1. Introduction

Deep generative modelling is a rapidly evolving area of research in which the goal is to model a complex probability distribution $p_{\mathbf{x}}^*(\mathbf{x})$ from a set of samples. Normalizing flows (NFs) in particular model an approximate density $p_{\mathbf{x}}(\mathbf{x})$ over the space $\mathcal{X}$ using a change-of-variables to a known base density $p_{\mathbf{z}}(\mathbf{z})$ over the space $\mathcal{Z}$. This change-of-variables, which is induced by a diffeomorphism $\mathbf{f} : \mathcal{Z} \to \mathcal{X}$, is given by

$$p_{\mathbf{x}}(\mathbf{x}) = p_{\mathbf{z}}\left(\mathbf{f}^{-1}(\mathbf{x})\right)\left|\det \mathbf{J}_{\mathbf{f}}\left(\mathbf{f}^{-1}(\mathbf{x})\right)\right|^{-1}. \quad (1)$$

In this expression, $\mathbf{J}_{\mathbf{f}}(\mathbf{z})$ is the Jacobian matrix of $\mathbf{f}$ at the point $\mathbf{z}$. By parameterizing and composing classes of diffeomorphisms $\mathbf{f}_\theta$, a flexible change-of-variables map can be learned via maximum likelihood. Compared to other generative frameworks, NFs provide the unique combination of efficient inference, efficient sampling, and exact likelihood estimation.

One limiting aspect of NFs is that, since $\mathbf{f}$ must be a diffeomorphism, they can only describe a probability model with

---

[1]Layer 6 AI, Toronto, Ontario, Canada. Correspondence to: Brendan Leigh Ross <brendan@layer6.ai>, Jesse C. Cresswell <jesse@layer6.ai>.

Third workshop on *Invertible Neural Networks, Normalizing Flows, and Explicit Likelihood Models* (ICML 2021). Copyright 2021 by the author(s).

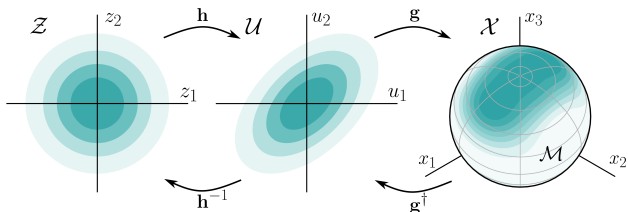

*Figure 1.* A normalized base density in the $\mathcal{Z}$ space is mapped by a bijective flow $\mathbf{h}$ to a more complicated density in $\mathcal{U}$. The injective component $\mathbf{g}$ maps this density onto a manifold $\mathcal{M}$ in $\mathcal{X}$.

full support. Conversely, many real-world datasets are assumed to exist on a low-dimensional submanifold $\mathcal{M} \subset \mathcal{X}$ (Fefferman et al., 2016). In the context of VAEs, Dai & Wipf (2019) observe that this situation encourages likelihood-based models to converge to infinity on $\mathcal{M}$ while ignoring the probability distribution of the data entirely. Behrmann et al. (2021) point out that invertible neural networks can become numerically non-invertible when the effective dimensionality of data and latents are mismatched. Correctly learning the data manifold $\mathcal{M}$ along with its density may circumvent these pathologies.

There is growing research interest in *injective flows*, which account for unknown manifold structure by incorporating a change in dimensionality between the base density and data space (Brehmer & Cranmer, 2020; Cunningham et al., 2020; Kumar et al., 2020; Kothari et al., 2021; Cunningham & Fiterau, 2021). However, leading injective flow models still suffer from drawbacks including intractable density estimation (Brehmer & Cranmer, 2020) and reliance on stochastic inverses (Cunningham et al., 2020).

In this paper we propose Conformal Embedding Flows (CEFs), a class of flows that use *conformal embeddings* to transform from low to high dimensions while maintaining invertibility and an efficiently computable density. We show how conformal embeddings can be used to learn a lower dimensional data manifold, and we combine them with powerful NF architectures for learning densities. The overall CEF paradigm preserves the advantages of standard NFs: efficient density estimation, sampling, and inference. We propose several types of conformal embedding that can be implemented as composable layers of a flow, including a new invertible layer, the orthogonal $k \times k$ convolution.

## 2. Background

Injective flows seek to learn an explicitly low-dimensional support by reducing the dimensionality of the latent space. The flow is modelled as a *smooth embedding*: an injective function which is diffeomorphic to its image. This case can be accommodated with a generalized change of variables formula for densities as follows (Gemici et al., 2016).

Let $\mathbf{g} : \mathcal{U} \to \mathcal{X}$ be a smooth embedding from a latent space $\mathcal{U}$ onto the data manifold $\mathcal{M} \subset \mathcal{X}$. That is, $\mathcal{M} = \mathbf{g}(\mathcal{U})$ is the range of $\mathbf{g}$. Accordingly, $\mathbf{g}$ has a left-inverse[1] $\mathbf{g}^\dagger : \mathcal{X} \to \mathcal{U}$ which is smooth on $\mathcal{M}$ and satisfies $\mathbf{g}^\dagger(\mathbf{g}(\mathbf{u})) = \mathbf{u}$ for all $\mathbf{u} \in \mathcal{U}$. Using the shorthand $\mathbf{u} = \mathbf{g}^\dagger(\mathbf{x})$, the generalized change of variables formula defined for $\mathbf{x} \in \mathcal{M}$ can be written (Lee, 2018)

$$p_\mathbf{x}(\mathbf{x}) = p_\mathbf{u}(\mathbf{u}) \left| \det \left[ \mathbf{J}_\mathbf{g}^T(\mathbf{u}) \mathbf{J}_\mathbf{g}(\mathbf{u}) \right] \right|^{-\frac{1}{2}} , \qquad (2)$$

for a base density $p_\mathbf{u}(\mathbf{u})$. As before, it is helpful to introduce a latent space $\mathcal{Z}$ of dimension $m$ along with a diffeomorphism $\mathbf{h} : \mathcal{Z} \to \mathcal{U}$ representing a bijective NF between $\mathcal{Z}$ and $\mathcal{U}$. Taking the composed injective transformation $\mathbf{g} \circ \mathbf{h}$ and applying the chain rule $\mathbf{J}_{\mathbf{g} \circ \mathbf{h}} = \mathbf{J}_\mathbf{g} \mathbf{J}_\mathbf{h}$ simplifies the determinant in Eq. (2) since $\mathbf{J}_\mathbf{h}$ is square: $\det \left[ \mathbf{J}_\mathbf{h}^T \mathbf{J}_\mathbf{g}^T \mathbf{J}_\mathbf{g} \mathbf{J}_\mathbf{h} \right] = (\det \mathbf{J}_\mathbf{h})^2 \det \left[ \mathbf{J}_\mathbf{g}^T \mathbf{J}_\mathbf{g} \right]$. Writing $\mathbf{z} = \mathbf{h}^{-1}(\mathbf{u})$, the data density is modelled by

$$p_\mathbf{x}(\mathbf{x}) = p_\mathbf{z}(\mathbf{z}) \left| \det \mathbf{J}_\mathbf{h}(\mathbf{z}) \right|^{-1} \left| \det \left[ \mathbf{J}_\mathbf{g}^T(\mathbf{u}) \mathbf{J}_\mathbf{g}(\mathbf{u}) \right] \right|^{-\frac{1}{2}} , \qquad (3)$$

with the entire process depicted in Fig. 1.

If the manifold is unknown, or if $g(\mathbf{u})$ cannot perfectly fit the data, points outside of $g(\mathcal{U})$ will arise during training. In this case the model's log-likelihood will be $-\infty$. Cunningham et al. (2020) remedy this by adding an off-manifold noise term to the model, but inference requires a stochastic inverse, and the model must be optimized using an ELBO-like objective. Other work (Brehmer & Cranmer, 2020; Kothari et al., 2021) has projected data to the manifold via $\mathbf{g} \circ \mathbf{g}^\dagger$ prior to computing log-likelihoods, and optimized $\mathbf{g}$ using the reconstruction loss $\mathbb{E}_{\mathbf{x} \sim p_\mathbf{x}^*} \|\mathbf{x} - \mathbf{g}(\mathbf{g}^\dagger(\mathbf{x}))\|^2$.

When computing likelihoods, the determinant term $\det \left[ \mathbf{J}_\mathbf{g}^T \mathbf{J}_\mathbf{g} \right]$ presents a computational challenge. Kumar et al. (2020) maximize it using an approximate lower bound, while Brehmer & Cranmer (2020) and Kothari et al. (2021) circumvent its computation altogether by only maximizing the other factors in the likelihood. In contrast to past injective flow models, our approach allows for straightforward evaluation and optimization of $\det \left[ \mathbf{J}_\mathbf{g}^T \mathbf{J}_\mathbf{g} \right]$ in the same way standard NFs do for $\det \mathbf{J}_\mathbf{f}$. As far as we can find, ours is the first approach to make this task tractable at scale.

For more related work, see App. A.

---

[1] † denotes a left-inverse function, not necessarily the matrix pseudoinverse.

## 3. Conformal Embedding Flows

In this section, we propose Conformal Embedding Flows (CEFs) as a method for learning both the low-dimensional manifold $\mathcal{M} \subset \mathcal{X}$ and the probability density of the data on the manifold.

Modern bijective flow work has produced tractable $\det \mathbf{J}_\mathbf{f}$ terms by designing layers with triangular Jacobians (Dinh et al., 2014; 2017). For injective flows, the combination $\mathbf{J}_\mathbf{g}^T \mathbf{J}_\mathbf{g}$ is symmetric, so it is triangular if and only if it is diagonal, meaning $\mathbf{J}_\mathbf{g}$ has orthogonal columns. While this restriction is feasible for a single layer $\mathbf{g}$, it is not composable. If $\mathbf{g}_1$ and $\mathbf{g}_2$ are embeddings whose Jacobians have orthogonal columns, it need not follow that $\mathbf{J}_{\mathbf{g}_2 \circ \mathbf{g}_1}$ has orthogonal columns. Additionally, since the Jacobians are not square, $\det \left[ \mathbf{J}_{\mathbf{g}_1}^T \mathbf{J}_{\mathbf{g}_2}^T \mathbf{J}_{\mathbf{g}_2} \mathbf{J}_{\mathbf{g}_1} \right]$, the determinant in Eq. (2), cannot be factored into a product of individually computable terms. To ensure composability, we propose enforcing the slightly stricter criterion that each $\mathbf{J}_\mathbf{g}^T \mathbf{J}_\mathbf{g}$ be a scalar multiple of the identity matrix. This is precisely the condition that $\mathbf{g}$ is a conformal embedding.

Formally, a smooth embedding $\mathbf{g} : \mathcal{U} \to \mathcal{X}$ is a *conformal embedding* if its Jacobian satisfies

$$\mathbf{J}_\mathbf{g}^T(\mathbf{u}) \mathbf{J}_\mathbf{g}(\mathbf{u}) = \lambda^2(\mathbf{u}) \mathbf{I}_m , \qquad (4)$$

where $m$ is the dimensionality of $\mathcal{U}$ and $\lambda : \mathcal{U} \to \mathbb{R}$ is a smooth non-zero scalar function, the *conformal factor* (Lee, 2018). In other words, $\mathbf{J}_\mathbf{g}$ has orthonormal columns up to a non-zero multiplicative constant. If $\mathbf{h} : \mathcal{Z} \to \mathcal{U}$ is a standard flow model, the injective flow $\mathbf{g} \circ \mathbf{h} : \mathcal{Z} \to \mathcal{X}$ satisfies

$$p_\mathbf{x}(\mathbf{x}) = p_\mathbf{z}(\mathbf{z}) \left| \det \mathbf{J}_\mathbf{h}(\mathbf{z}) \right|^{-1} \lambda^{-m}(\mathbf{u}) . \qquad (5)$$

We call $\mathbf{g} \circ \mathbf{h}$ a Conformal Embedding Flow.

There are multiple options for training a CEF. End-to-end maximum likelihood training could be used for $\mathbf{g} \circ \mathbf{h}$ as a whole, but when projecting to the manifold, it is possible to maximize density without learning the manifold correctly (Brehmer & Cranmer, 2020). We find it effective to first train the manifold learner $\mathbf{g}$ alone for several epochs by minimizing the reconstruction loss $\mathbb{E}_{\mathbf{x} \sim p_\mathbf{x}^*} \|\mathbf{x} - \mathbf{g}(\mathbf{g}^\dagger(\mathbf{x}))\|^2$. After this manifold warmup phase, our model density can be optimized in two ways. The first is the *sequential* training approach, in which we optimize $\mathbf{h}$ with $\mathbf{g}$ fixed as in Brehmer & Cranmer (2020). The alternative, which we refer to as the *joint* training approach, is to optimize the following loss:

$$\mathcal{L} = \mathbb{E}_{\mathbf{x} \sim p_\mathbf{x}^*} \left[ -\log p_\mathbf{x}(\mathbf{x}) + \alpha \|\mathbf{x} - \mathbf{g}(\mathbf{g}^\dagger(\mathbf{x}))\|^2 \right] . \qquad (6)$$

This mixed loss is unique to our model because it is the first model for which $\log p_\mathbf{x}(\mathbf{x})$ is tractable in its entirety.

## 3.1. Designing Conformal Embedding Flows

To build a parameterizable and scalable model $\mathbf{g}$, it helps to work with conformal building blocks $\mathbf{g}_i : \mathcal{U}_{i-1} \to \mathcal{U}_i$ (where $\mathcal{U}_0 = \mathcal{U}$ and $\mathcal{U}_k = \mathcal{X}$), which we compose to produce the full conformal embedding $\mathbf{g} = \mathbf{g}_k \circ \cdots \circ \mathbf{g}_1$. In turn, $\mathbf{g}$ is conformal because

$$\mathbf{J}_\mathbf{g}^T \mathbf{J}_\mathbf{g} = \left( \mathbf{J}_{\mathbf{g}_1}^T \cdots \mathbf{J}_{\mathbf{g}_k}^T \right) \left( \mathbf{J}_{\mathbf{g}_k} \cdots \mathbf{J}_\mathbf{g} \right) = \lambda_1^2 \cdots \lambda_k^2 \mathbf{I}_m \ . \quad (7)$$

Our goal in the remainder of this section is to design classes of conformal building blocks which can be parameterized and learned in a CEF.

### 3.1.1. CONFORMAL EMBEDDINGS FROM CONFORMAL MAPPINGS

Consider the special case where the conformal embedding maps between Euclidean spaces $\mathcal{U} \subseteq \mathbb{R}^d$ and $\mathcal{V} \subseteq \mathbb{R}^d$ of the same dimension. Liouville's theorem (Hartman, 1958) states that any such *conformal mapping* can be expressed as a composition of translations, orthogonal transformations, scalings, and inversions, all of which are defined in Table 1. We created conformal embeddings primarily by composing these layers. Zero-padding (which is trivially conformal) was interspersed to provide changes in dimensionality (Brehmer & Cranmer, 2020).

*Table 1.* Conformal Mappings

| Type | Functional Form | Inverse | $\lambda(\mathbf{u})$ |
|---|---|---|---|
| Translation | $\mathbf{u} \mapsto \mathbf{u} + \mathbf{a}$ | $\mathbf{v} \mapsto \mathbf{v} - \mathbf{a}$ | 1 |
| Orthogonal | $\mathbf{u} \mapsto \mathbf{Qu}, \quad \mathbf{Q} \in O(d)$ | $\mathbf{v} \mapsto \mathbf{Q}^T \mathbf{v}$ | 1 |
| Scaling | $\mathbf{u} \mapsto \lambda \mathbf{u}$ | $\mathbf{v} \mapsto \lambda^{-1} \mathbf{v}$ | $\lambda$ |
| Inversion | $\mathbf{u} \mapsto \mathbf{u}/\|\mathbf{u}\|^2$ | $\mathbf{v} \mapsto \mathbf{v}/\|\mathbf{v}\|^2$ | $\|\mathbf{u}\|^{-2}$ |

We parameterized orthogonal transformations in two different ways: as Householder matrices (Tomczak & Welling, 2016) and as the matrix exponential of a skew-symmetric matrix using (Lezcano-Casado, 2019).

To scale orthogonal transformations to image data, we propose a new invertible layer: the orthogonal $k \times k$ convolution. In the spirit of the invertible $1 \times 1$ convolutions of Kingma & Dhariwal (2018), we note that a $k \times k$ convolution with stride $k$ has a block diagonal Jacobian. It suffices then to constrain the filters such that these blocks are orthogonal. These layers can be inverted efficiently by applying a transposed convolution with the same filters, while a standard invertible $1 \times 1$ convolution requires a matrix inversion. This facilitates quick forward and inverse passes when performing reconstructions during training.

### 3.1.2. PIECEWISE CONFORMAL EMBEDDINGS

To increase expressivity of the embeddings, the conformality condition on $\mathbf{g}$ can be relaxed to the point of being

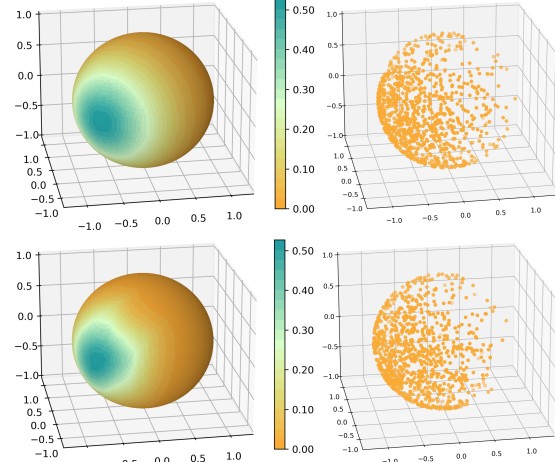

*Figure 2.* (Top) A density $p_\mathbf{x}^*(\mathbf{x})$ with support on the sphere, and $10^3$ samples comprising the training dataset $\{\mathbf{x_i}\}$. (Bottom) The density learned by a CEF, and $10^3$ generated samples.

conformal *almost everywhere*. This is similar to how the diffeomorphism property of standard flows is relaxed when rectifier nonlinearities are used in coupling layers (Dinh et al., 2017). We considered the following two piecewise conformal embeddings. *Conformal ReLU*, based on the injective ReLU proposed by Kothari et al. (2021), is defined by the following forward ($\mathcal{U} \to \mathcal{V}$) and left-inverse ($\mathcal{V} \to \mathcal{U}$) functions with $\mathbf{Q} \in O(d)$:

$$\mathbf{u} \mapsto \text{ReLU} \begin{bmatrix} \mathbf{Qu} \\ -\mathbf{Qu} \end{bmatrix}, \quad \begin{bmatrix} \mathbf{v}_1 \\ \mathbf{v}_2 \end{bmatrix} \mapsto \mathbf{Q}^T (\mathbf{v}_1 - \mathbf{v}_2), \quad (8)$$

and $\lambda(\mathbf{u}) = 1$. We believe it to be of general interest as a conformal nonlinearity, but it provided no performance improvements in our experiments. More useful was the *Conditional Orthogonal* transformation,

$$\mathbf{u} \mapsto \begin{cases} \mathbf{Q}_1 \mathbf{u} & \text{if } \|\mathbf{u}\| < 1, \\ \mathbf{Q}_2 \mathbf{u} & \text{if } \|\mathbf{u}\| \geq 1, \end{cases} \quad \mathbf{v} \mapsto \begin{cases} \mathbf{Q}_1^T \mathbf{u} & \text{if } \|\mathbf{v}\| < 1, \\ \mathbf{Q}_2^T \mathbf{u} & \text{if } \|\mathbf{v}\| \geq 1, \end{cases}$$
$$(9)$$

with $\mathbf{Q}_1, \mathbf{Q}_2 \in O(d)$, and $\lambda(\mathbf{u}) = 1$, which takes advantage of the norm-preservation of orthogonal transformations to create an invertible layer. Despite the conditional orthogonal layer being discontinuous, it provided a boost in reconstruction ability on image data.

## 4. Experiments

### 4.1. Synthetic Spherical Data

To demonstrate how a CEF can jointly learn a manifold and density, we generated a synthetic dataset from a known distribution with support on a spherical surface embedded in $\mathbb{R}^3$. This ground truth density is visualized in Fig. 2. We constructed the conformal embedding $\mathbf{g}$ from a padding

*Table 2.* CelebA Performance

| MODEL | FID | RECON | g + h PARAMETERS |
|-------|-----|-------|------------------|
| S-MF  | $110.9 \pm 0.1$ | $0.40 \times 10^{-3}$ | $36346752 + 36429778$ |
| S-CEF | $111.8 \pm 0.1$ | $1.05 \times 10^{-3}$ | $83036 + 36429778$ |
| J-CEF | $126.0 \pm 0.2$ | $1.04 \times 10^{-3}$ | $83036 + 36429778$ |

layer and the conformal mappings in Table 1, and the bijective component **h** consisted of three Glow-style steps acting on the two latent dimensions. We trained the two components of the CEF jointly with the mixed loss function in Eq. (6). The resulting model density is plotted in Fig. 2 along with generated samples, and it shows good fidelity to the known manifold and density. See App. C.1 for complete details.

### 4.2. Image Modelling

We scale CEFs to larger data by training on the CelebA dataset, for which a low-dimensional manifold structure is postulated but not known (Liu et al., 2015). Our aim is to show that, although their functional form is restricted, CEFs are competitive with mainstream injective flow training approaches. In doing so, we are the first to perform end-to-end maximum likelihood training with an injective flow on image data. Three approaches were evaluated: a baseline sequentially trained manifold-flow (S-MF) (Brehmer & Cranmer, 2020), a sequentially trained CEF (S-CEF), and a jointly trained CEF (J-CEF). We fixed a small Glow-style stump with 3 levels and 3 steps per level as the backbone **h** for all models. See App. C.2 for further details.

Injective models cannot be compared on the basis of log-likelihood, since each model may have a different manifold support. Instead, we compare generated images on the basis of FID score, a measure of distance between image features that correlates well with human perception of image quality (Heusel et al., 2017). Lower FID scores are better. As expected, since the baseline embedding is much larger and more flexible, it achieves smaller reconstruction losses than the conformal models. However, the CEFs produce competitive FID scores, possibly due to the better global consistency of facial features from the two CEF models.

## 5. Conclusion

In this paper, we introduced Conformal Embedding Flows as a new framework for modelling probability distributions on low-dimensional manifolds while maintaining tractable densities. We showed that conformal embeddings improve upon past injective flow models by providing the unique combination of fast sampling, invertibility for inference, and a simple Jacobian determinant factor for efficient likelihood estimation, all while being composable for scaling

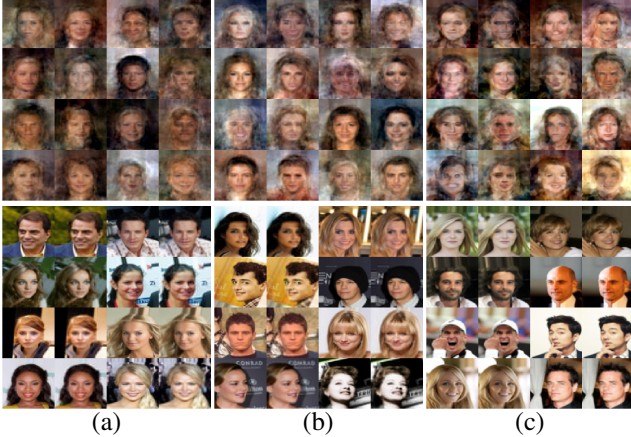

(a)       (b)       (c)

*Figure 3.* (Top) Generated samples and (Bottom) reconstructions for (a) J-CEF, (b) S-CEF, and (c) S-MF. Original images are on the left, and reconstructions are on the right.

to deep models. Furthermore, the restrictions on conformal embeddings are minimal, in that any looser condition will sacrifice one or more of these properties.

Strictly manifold-supported probability models such as ours introduce a bi-objective optimization problem. How to balance these objectives is unclear and, thus far, empirical (Brehmer & Cranmer, 2020). The difference in supports between two manifold models also makes their likelihoods incomparable. Cunningham et al. (2020) have made progress in this direction by adding noise to data generated on the manifold, but this makes inversion stochastic and introduces density estimation challenges. We suspect that using conformal manifold-learners may make density estimation more tractable in this setting, but further research is needed in this direction.

Just as standard flows trade some expressibility for tractable likelihoods, so must injective flows. Our conformal networks in particular are less expressive than state-of-the-art flow models. We mostly compose padding layers and dimension-*preserving* conformal mappings, which is a naturally restrictive class by Liouville's theorem (Hartman, 1958). Just as early work on NFs (Rezende & Mohamed, 2015; Dinh et al., 2014) introduced limited classes of parameterizable bijections, which were later improved in many directions, our work introduces several classes of parameterizable conformal embeddings. We expect that future work will uncover more expressive dimension-*increasing* conformal embeddings.

## Acknowledgements

We thank Gabriel Loaiza-Ganem and Anthony Caterini for their valuable discussions and advice. We also thank Parsa Torabian for sharing his experience with orthogonal weights and Maksims Volkovs for his helpful feedback.

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

## A. Related Work

**Flows on prescribed manifolds** Flows have been developed for Riemannian manifolds $\mathcal{M} \subseteq \mathcal{X}$ which are known in advance and can be defined as the image of some fixed $\phi : \mathcal{U} \rightarrow \mathcal{X}$, where $\mathcal{U} \subseteq \mathbb{R}^m$ (Gemici et al., 2016; Mathieu & Nickel, 2020; Papamakarios et al., 2021). In particular, Rezende et al. (2020) model densities on spheres and tori with convex combinations of Möbius transformations, which are cognate to conformal mappings. For known manifolds $\phi$ is fixed, and the density's log Jacobian determinant term may be computable in closed form. Our work replaces $\phi$ with a trainable network $g$, but the log Jacobian determinant still has a simple closed form.

**Flows on learnable manifolds** Extending flows to learnable manifolds brings about two main challenges: handling off-manifold points, and training the density on the manifold.

If the manifold is the density's support, all off-manifold points during training will have a likelihood of 0. This has been addressed by adding an off-manifold noise term (Cunningham et al., 2020; Cunningham & Fiterau, 2021) or by training the manifold with a reconstruction loss and projecting the data onto the manifold (Kumar et al., 2020; Brehmer & Cranmer, 2020; Kothari et al., 2021). We opt for the latter approach.

Training the density on the manifold is challenging because the log-determinant term is typically intractable. Kumar et al. (2020) use an approximate lower bound to train the log-determinant, while Brehmer & Cranmer (2020) and Kothari et al. (2021) separate the flow into two components, and only train the low-dimensional part of the flow. Our approach is the first injective model to provide a learnable manifold with exact log-determinant computation.

**Conformal, isometric, and orthogonal networks** Conformality has two special cases of interest: isometry and semi-orthogonality. An isometric embedding is a conformal embedding with a constant conformal factor of 1, while a semi-orthogonal transformation is a linear isometric embedding. Authors have typically imposed conformality (Peterfreund et al., 2020), isometry (Qi et al., 2020; Xiao et al., 2018), or semi-orthogonality (Bansal et al., 2018; Jia et al., 2017) as a regularizer. These approaches have not enforced orthogonality strictly; the set of $n \times k$ semi-orthogonal matrices forms the *Stiefel manifold* in $\mathbb{R}^{n \times k}$, so strict orthogonality entails optimizing weights along this manifold.

Past work has trained along the Stiefel manifold in two ways: with Riemannian gradient descent, and by directly parametrizing the manifold. Riemannian gradient descent algorithms typically require a singular value decomposition or QR decomposition at each training step (Harandi & Fernando, 2016; Ozay & Okatani, 2016; Huang et al., 2018). On the other hand, Lezcano-Casado (2019) and Lezcano-Casado & Martínez-Rubio (2019) parameterize the set of orthogonal matrices with positive determinant as the matrix exponential of a skew-symmetric matrix, wherein the matrix exponential and its gradient must be approximated. Another approach is to directly parameterize a smaller subset of orthogonal matrices. Tomczak & Welling (2016) construct a linear volume-preserving flow using $n \times n$ Householder matrices, which can be parameterized with $n - 1$ degrees of freedom. We found the matrix exponential and Householder approaches to be most efficient, so we used a mix of them.

## B. Details on Conformal Embeddings and Conformal Mappings

Let $(\mathcal{U}, \eta_{\mathbf{u}})$ and $(\mathcal{X}, \eta_{\mathbf{x}})$ be two Riemannian manifolds. We define a diffeomorphism $\mathbf{f} : \mathcal{U} \to \mathcal{X}$ to be a *conformal diffeomorphism* if it pulls back the metric $\eta_{\mathbf{x}}$ to some non-zero scalar multiple of $\eta_{\mathbf{u}}$ (Lee, 2018). That is,

$$\mathbf{f}^* \eta_{\mathbf{x}} = \lambda \eta_{\mathbf{u}} \tag{10}$$

for some smooth scalar function $\lambda \neq 0$. Many authors require $\lambda$ to be positive, but we allow it to be negative as well. Furthermore, we define a smooth embedding $\mathbf{g} : \mathcal{U} \to \mathcal{X}$ to be a *conformal embedding* if it is a conformal diffeomorphism onto its image $(\mathbf{g}(\mathcal{U}), \eta_{\mathbf{x}})$, where $\eta_{\mathbf{x}}$ is inherited from the ambient space $\mathcal{X}$.

In our context, $\mathcal{U} \subseteq \mathbb{R}^m$, $\mathcal{X} = \mathbb{R}^n$, and $\eta_{\mathbf{u}}$ and $\eta_{\mathbf{x}}$ are Euclidean metrics. This leads to an equivalent property (Eq. (4)):

$$\mathbf{J}_{\mathbf{g}}^T(\mathbf{u})\mathbf{J}_{\mathbf{g}}(\mathbf{u}) = \lambda^2(\mathbf{u})\mathbf{I}_m. \tag{11}$$

This also guarantees that $\det[\mathbf{J}_{\mathbf{g}}^T \mathbf{J}_{\mathbf{g}}] = \lambda^{2m}$ is tractable, even when $\mathbf{g} = \mathbf{g}_k \circ ... \circ \mathbf{g}_1$ is composed from several layers, as is needed for scalable injective flows.

To demonstrate that conformal embeddings are an expressive class of functions, we first turn to the most restricted case where $n = m$; i.e. conformal mappings. In Apps. B.1 and B.2 we provide an intuitive investigation of the classes of conformal mappings using infinitesimals. We then discuss in App. B.3 why conformal embeddings in general are more challenging to analyze, but also show intuitively why they are more expressive than dimension-preserving conformal mappings.

### B.1. Infinitesimal Conformal Mappings

Consider a mapping of Euclidean space with dimension $m \geq 3$. Liouville's theorem for conformal mappings constrains the set of such maps which satisfy the conformal condition Eq. (11). Such functions can be decomposed into translations, orthogonal transformations, scalings, and inversions. Here we provide a direct approach for the interested reader, which also leads to some insight on the general case of conformal embeddings (Di Francesco et al., 2012). First we will find all infinitesimal transformations which satisfy the conformal condition, then exponentiate them to obtain the set of finite conformal mappings.

Consider a transformation $\mathbf{f} : \mathbb{R}^m \to \mathbb{R}^m$ which is infinitesimally close to the identity function, expressed in Cartesian coordinates as

$$\mathbf{f}(\mathbf{x}) = \mathbf{x} + \boldsymbol{\epsilon}(\mathbf{x}). \tag{12}$$

That is, we only keep terms linear in the infinitesimal quantity $\boldsymbol{\epsilon}$. The mappings produced will only encompass transformations which are continuously connected to the identity,

but we restrict our attention to these for now. However, this simple form allows us to directly study how Eq. (11) constrains the infinitesimal $\boldsymbol{\epsilon}(\mathbf{x})$:

$$\mathbf{J}_{\mathbf{f}}^T(\mathbf{x})\mathbf{J}_{\mathbf{f}}(\mathbf{x}) = \left[\mathbf{I}_m + \frac{\partial \boldsymbol{\epsilon}}{\partial \mathbf{x}}\right]^T \left[\mathbf{I}_m + \frac{\partial \boldsymbol{\epsilon}}{\partial \mathbf{x}}\right]$$
$$= \mathbf{I}_m + \frac{\partial \boldsymbol{\epsilon}}{\partial \mathbf{x}}^T + \frac{\partial \boldsymbol{\epsilon}}{\partial \mathbf{x}}. \tag{13}$$

By Eq. (11), the symmetric sum of $\partial \boldsymbol{\epsilon}/\partial \mathbf{x}$ must be proportional to the identity matrix. Let us call the position-dependent proportionality factor $\eta(\mathbf{x})$. We can start to understand $\eta(\mathbf{x})$ by taking a trace

$$\frac{\partial \boldsymbol{\epsilon}}{\partial \mathbf{x}}^T + \frac{\partial \boldsymbol{\epsilon}}{\partial \mathbf{x}} = \eta(\mathbf{x})\mathbf{I}_m, \tag{14}$$

$$\frac{2}{m}\text{tr}\left(\frac{\partial \boldsymbol{\epsilon}}{\partial \mathbf{x}}\right) = \eta(\mathbf{x}). \tag{15}$$

Taking another derivative of Eq. (14) proves to be useful, so we switch to index notation to handle the tensor multiplications,

$$\frac{\partial}{\partial x_k}\frac{\partial \epsilon_j}{\partial x_i} + \frac{\partial}{\partial x_k}\frac{\partial \epsilon_i}{\partial x_j} = \frac{\partial \eta}{\partial x_k}\delta_{ij}, \tag{16}$$

where the Kronecker delta $\delta_{ij}$ is 1 if $i = j$, and 0 otherwise. On the left-hand-side, derivatives can be commuted. By taking a linear combination of the three permutations of indices we come to

$$2\frac{\partial}{\partial x_k}\frac{\partial \epsilon_i}{\partial x_j} = \frac{\partial \eta}{\partial x_j}\delta_{ik} + \frac{\partial \eta}{\partial x_k}\delta_{ij} - \frac{\partial \eta}{\partial x_i}\delta_{jk}. \tag{17}$$

Summing over elements where $j = k$ gives the Laplacian of $\epsilon_i$, while picking up only the derivatives of $\eta$ with respect to $x_i$, so we can switch back to vector notation where

$$2\nabla^2\boldsymbol{\epsilon} = (2 - m)\frac{\partial \eta}{\partial \mathbf{x}}. \tag{18}$$

Now we have two equations (14) and (18)[2] involving derivatives of $\boldsymbol{\epsilon}$ and $\eta$. To eliminate $\boldsymbol{\epsilon}$, we can apply $\nabla^2$ to (14), while applying $\partial/\partial \mathbf{x}$ to (18)

$$\nabla^2\frac{\partial \boldsymbol{\epsilon}}{\partial \mathbf{x}}^T + \nabla^2\frac{\partial \boldsymbol{\epsilon}}{\partial \mathbf{x}} = \nabla^2\eta\mathbf{I}_d \tag{19}$$

$$2\nabla^2\frac{\partial \boldsymbol{\epsilon}}{\partial \mathbf{x}} = (2 - m)\frac{\partial^2 \eta}{\partial \mathbf{x}\partial \mathbf{x}}. \tag{20}$$

Since Eq. (20) is manifestly symmetric, the left-hand-sides are actually equal. Equating the right-hand-sides, we can again sum the diagonal terms, giving the much simpler form

$$(m - 1)\nabla^2\eta = 0. \tag{21}$$

---

[2] We note that the steps following Eq. (18) are only justified for $m \geq 3$ which we have assumed. In two dimensions the conformal group is much larger and Liouville's theorem no longer captures all conformal mappings.

Ultimately, revisiting Eq. (20) shows that the function $\eta(\mathbf{x})$ is linear in the coordinates

$$\frac{\partial^2 \eta}{\partial \mathbf{x} \partial \mathbf{x}} = 0 \implies \eta(\mathbf{x}) = \alpha + \boldsymbol{\beta} \cdot \mathbf{x}, \quad (22)$$

for constants $\alpha, \boldsymbol{\beta}$. This allows us to relate back to the quantity of interest $\boldsymbol{\epsilon}$. Skimming back over the results so far, the most general equation where having the linear expression for $\eta(\mathbf{x})$ helps is Eq. (17) which now is

$$2\frac{\partial}{\partial x_k}\frac{\partial \epsilon_i}{\partial x_j} = \beta_j \delta_{ik} + \beta_k \delta_{ij} - \beta_i \delta_{jk}. \quad (23)$$

The point is that the right-hand-side is constant, meaning that $\boldsymbol{\epsilon}(\mathbf{x})$ is at most quadratic in $\mathbf{x}$. Hence, we can make an ansatz for $\boldsymbol{\epsilon}$ in full generality, involving sets of infinitesimal constants

$$\boldsymbol{\epsilon} = \mathbf{a} + \mathbf{B}\mathbf{x} + \mathbf{x}\overset{\leftrightarrow}{\mathbf{C}}\mathbf{x}, \quad (24)$$

where $\overset{\leftrightarrow}{\mathbf{C}} \in \mathbb{R}^{m \times m \times m}$ is a 3-tensor.

So far we have found that infinitesimal conformal transformations can have at most quadratic dependence on the coordinates. It remains to determine the constraints on each set of constants $\mathbf{a}$, $\mathbf{B}$, and $\overset{\leftrightarrow}{\mathbf{C}}$, and interpret the corresponding mappings. We consider each of them in turn.

All constraints on $\boldsymbol{\epsilon}$ involve derivatives, so there is nothing more to say about the constant term. It represents an infinitesimal translation

$$\mathbf{f}(\mathbf{x}) = \mathbf{x} + \mathbf{a}. \quad (25)$$

On the other hand, the linear term is constrained by Eqs. (14) and (15) which give

$$\mathbf{B} + \mathbf{B}^T = \frac{2}{m}\text{tr}(\mathbf{B})\mathbf{I}_m. \quad (26)$$

Hence, $\mathbf{B}$ has an unconstrained anti-symmetric part $\mathbf{B}_{\text{AS}} = \frac{1}{2}(\mathbf{B} - \mathbf{B}^T)$ representing an infinitesimal rotation

$$\mathbf{f}(\mathbf{x}) = \mathbf{x} + \mathbf{B}_{\text{AS}}\mathbf{x}, \quad (27)$$

while its symmetric part is diagonal as in Eq. (26),

$$\mathbf{f}(\mathbf{x}) = \mathbf{x} + \lambda\mathbf{x}, \quad \lambda = \frac{1}{m}\text{tr}(\mathbf{B}), \quad (28)$$

which is an infinitesimal scaling. This leaves only the quadratic term for interpretation which is more easily handled in index notation, i.e. $\epsilon_i = \sum_{lm} C_{ilm} x_l x_m$. The quadratic term is significantly restricted by Eq. (23),

$$2\frac{\partial^2}{\partial x_k \partial x_j}\sum_{lm} C_{ilm} x_l x_m = 2C_{ijk}$$
$$= \beta_j \delta_{ik} + \beta_k \delta_{ij} - \beta_i \delta_{jk}. \quad (29)$$

This allows us to isolate $\beta_k$ in terms of $C_{ijk}$, specifically from the trace over $C$'s first two indices,

$$2\sum_{i=j} C_{ijk} = \beta_k + \beta_k m - \beta_k = \beta_k m. \quad (30)$$

Hereafter we use $b_k = \beta_k/2 = \sum_{i=j} C_{ijk}/m$. Then with Eq. (29) the corresponding infinitesimal transformation is

$$f_i(\mathbf{x}) = x_i + \sum_{jk} C_{ijk} x_j x_k$$
$$= x_i + \sum_{jk}(b_j \delta_{ik} + b_k \delta_{ij} - b_i \delta_{jk}) x_j x_k$$
$$= x_i + 2x_i \sum_j b_j x_j - b_i \sum_j (x_j)^2, \quad (31)$$
$$\mathbf{f}(\mathbf{x}) = \mathbf{x} + 2(\mathbf{b} \cdot \mathbf{x})\mathbf{x} - \|\mathbf{x}\|^2 \mathbf{b}.$$

We postpone the interpretation momentarily.

Thus we have found all continuously parameterizable infinitesimal conformal mappings connected to the identity, and showed they come in four distinct types. By composing infinitely many such transformations, or "exponentiating" them, we obtain finite conformal mappings. Formally, this is the process of exponentiating the elements of a Lie algebra to obtain elements of a corresponding Lie group.

### B.2. Finite Conformal Mappings

As an example of obtaining finite mappings from infinitesimal ones we take the infinitesimal rotations from Eq. (27) where we note that $\mathbf{f}$ only deviates from the identity by an infinitesimal vector field $\mathbf{B}_{\text{AS}}\mathbf{x}$. By integrating the field we get the finite displacement of any point under many applications of $\mathbf{f}$, i.e. the integral curves $\mathbf{x}(t)$ defined by

$$\dot{\mathbf{x}}(t) = \mathbf{B}_{\text{AS}}\mathbf{x}(t), \quad \mathbf{x}(0) = \mathbf{x}_0. \quad (32)$$

This differential equation has the simple solution

$$\mathbf{x}(t) = \exp(t\mathbf{B}_{\text{AS}})\mathbf{x}_0. \quad (33)$$

Finally we recognize that when a matrix $\mathbf{A}$ is antisymmetric, the matrix exponential $e^{\mathbf{A}}$ is orthogonal, showing that the finite transformation given by $t = 1$, $\mathbf{f}(\mathbf{x}_0) = \exp(\mathbf{B}_{\text{AS}})\mathbf{x}_0$, is indeed a rotation. Furthermore, it is intuitive that infinitesimal translations and scalings also compose into finite translations and scalings. Examples are shown in Fig. 4 (a-c)

The infinitesimal transformation in Eq. (31) is non-linear in $\mathbf{x}$, so it does not exponentiate easily as for the other three cases. It helps to linearize with a change of coordinates $\mathbf{y} = \mathbf{x}/\|\mathbf{x}\|^2$ which happens to be an inversion:

$$\dot{\mathbf{x}}(t) = 2(\mathbf{b} \cdot \mathbf{x})\mathbf{x} - \|\mathbf{x}\|^2 \mathbf{b}, \quad (34)$$

$$\dot{\mathbf{y}}(t) = \frac{\dot{\mathbf{x}}}{\|\mathbf{x}\|^2} - 2\frac{\mathbf{x} \cdot \dot{\mathbf{x}}}{\|\mathbf{x}\|^4}\mathbf{x} = -\mathbf{b}. \quad (35)$$

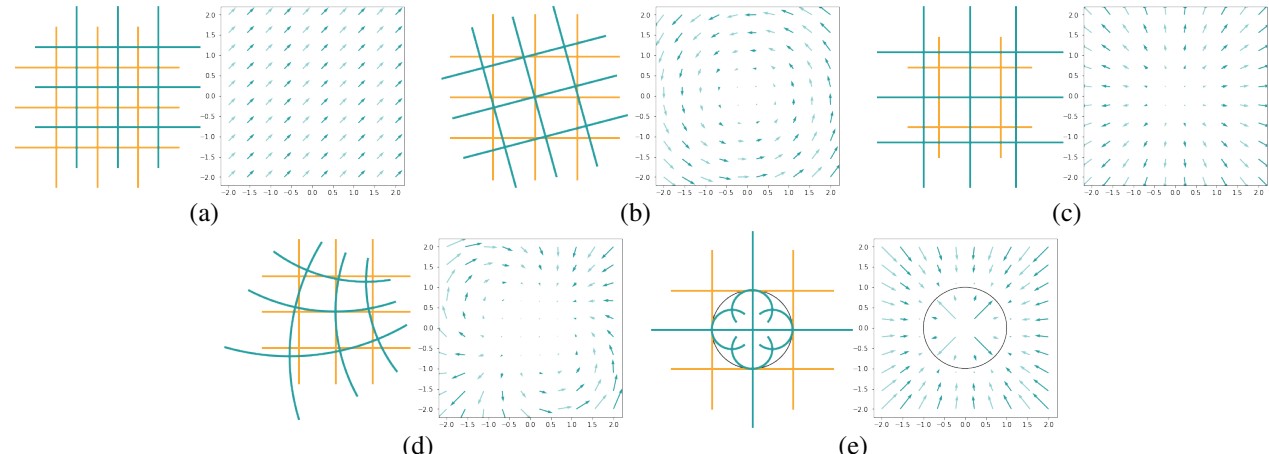

*Figure 4.* Effects of conformal mappings on gridlines, and their corresponding vector fields showing local displacements. Mappings are: (a) translation by $\mathbf{a} = [1, 1]$; (b) orthogonal transformation (2D rotation) by angle $\theta = \pi/12$; (c) scaling by $\lambda = 1.5$; (d) SCT by $\mathbf{b} = [-0.1, -0.1]$; (e) inversion, also showing the unit circle. The interior of the circle is mapped to the exterior, and vice versa.

We now get the incredibly simple solution $\mathbf{y}(t) = \mathbf{y}_0 - t\mathbf{b}$, a translation, after which we can undo the inversion

$$\frac{\mathbf{x}(t)}{\|\mathbf{x}\|^2} = \frac{\mathbf{x}_0}{\|\mathbf{x}_0\|^2} - t\mathbf{b}. \qquad (36)$$

This form is equivalent to a Special Conformal Transformation (SCT) (Di Francesco et al., 2012), which we can see by defining the finite transformation as $\mathbf{f}(\mathbf{x}_0) = \mathbf{x}(1)$, and taking the inner product of both sides with themselves

$$\|\mathbf{f}(\mathbf{x}_0)\|^2 = \frac{\|\mathbf{x}_0\|^2}{1 - 2\mathbf{b} \cdot \mathbf{x}_0 + \|\mathbf{b}\|^2\|\mathbf{x}_0\|^2}, \qquad (37)$$

and finally isolating

$$\begin{aligned}\mathbf{f}(\mathbf{x}_0) &= \frac{\|\mathbf{f}(\mathbf{x}_0)\|^2}{\|\mathbf{x}_0\|^2}\mathbf{x}_0 - \|\mathbf{f}(\mathbf{x}_0)\|^2\mathbf{b} \\ &= \frac{\mathbf{x}_0 - \|\mathbf{x}_0\|^2\mathbf{b}}{1 - 2\mathbf{b} \cdot \mathbf{x}_0 + \|\mathbf{b}\|^2\|\mathbf{x}_0\|^2}.\end{aligned} \qquad (38)$$

An example SCT is shown in Fig. 4 (d), demonstrating their non-linear nature. In the process of this derivation we have learned that SCTs can be interpreted as an inversion, followed by a translation by $-\mathbf{b}$, followed by an inversion, and the infinitesimal Eq. (31) is recovered when the translation is small.

By composition, the four types of finite conformal mapping we have encountered, namely translations, rotations, scalings, and SCTs, generate the conformal group - the group of transformations of Euclidean space which locally preserve angles and orientation. The infinitesimal transformations we derived directly give the corresponding elements of the Lie algebra.

Eq. (11) also admits non-orientation preserving solutions which are not generated by the infinitesimal approach. Composing the scalings in Eq. (28) only produces finite scalings

by a positive factor, i.e. $\mathbf{f}(\mathbf{x}) = e^\lambda \mathbf{x}$. Similarly, composing infinitesimal rotations does not generate reflections - non-orientation preserving orthogonal transformations that are not connected to the identity. The conformal group can be extended by including non-orientation preserving transformations, namely inversions (Fig. 4 (e)), negative scalings, and reflections as in Table 1. All of these elements still satisfy Eq. (11), as do their closure under composition. By Liouville's theorem, these comprise all possible conformal mappings.

The important point for our discussion is that any conformal mapping can be built up from the simple elements in Table 1. In other words, a neural network can learn any conformal mapping by learning a sequence of the simple elements.

**B.3. Conformal Embeddings**

Whereas conformal mappings have been exhaustively classified, conformal embeddings have not. While the defining equations for a conformal embedding $\mathbf{g} : \mathcal{U} \to \mathcal{X}$, namely

$$\mathbf{J}_{\mathbf{g}}^T(\mathbf{u})\mathbf{J}_{\mathbf{g}}(\mathbf{u}) = \lambda^2(\mathbf{u})\mathbf{I}_m, \qquad (39)$$

appear similar to those of conformal mappings, we cannot apply the techniques from Apps. B.1 and B.2 to enumerate them. Conformal embeddings do not necessarily have identical domain and codomain. As such, finite conformal embeddings can not be generated by exponentiating infinitesimals.

The lack of full characterization of all conformal embeddings hints that they are a much richer class of functions. For a more concrete understanding, we can study Eq. (39) as a system of PDEs. This system consists of $m(m + 1)/2$ independent equations (noting the symmetry of $\mathbf{J}_{\mathbf{g}}^T\mathbf{J}_{\mathbf{g}}$) to be satisfied by $n + 1$ functions, namely $\mathbf{g}(\mathbf{u})$ and $\lambda(\mathbf{u})$. In

the typical case that $n < m(m+1)/2 - 1$, i.e. $n$ is not significantly larger than $m$, the system is overdetermined. Despite this, solutions do exist. We have already seen that the most restricted case $n = m$ of conformal mappings admits four qualitatively different classes of solutions. These remain solutions when $n > m$ simply by having $\mathbf{g}$ map to a constant in the extra $n - m$ dimensions.

Intuitively, adding an extra dimension for solving the PDEs is similar to introducing a slack variable in an optimization problem. In case it is not clear that adding additional functions $\mathbf{g}_i, i > m$ enlarges the class of solutions of Eq. (39), we provide a concrete example. Take the case $n = m = 2$ for a fixed $\lambda(u_1, u_2)$. The system of equations that $\mathbf{g}(\mathbf{u})$ must solve is

$$\left(\frac{\partial g_1}{\partial u_1}\right)^2 + \left(\frac{\partial g_2}{\partial u_1}\right)^2 = \lambda^2(u_1, u_2),$$

$$\left(\frac{\partial g_1}{\partial u_2}\right)^2 + \left(\frac{\partial g_2}{\partial u_2}\right)^2 = \lambda^2(u_1, u_2), \qquad (40)$$

$$\frac{\partial g_1}{\partial u_1}\frac{\partial g_1}{\partial u_2} + \frac{\partial g_2}{\partial u_1}\frac{\partial g_2}{\partial u_2} = 0.$$

Suppose that for the given $\lambda(u_1, u_2)$ no complete solution exists, but we do have a $\mathbf{g}(\mathbf{u})$ which simultaneously solves all but the first equation. Enlarging the codomain $\mathcal{X}$ with an additional dimension ($n = 3$) gives an additional function $g_3(\mathbf{u})$ to work with, while $\lambda(u_1, u_2)$ is unchanged. The system of equations becomes

$$\left(\frac{\partial g_1}{\partial u_1}\right)^2 + \left(\frac{\partial g_2}{\partial u_1}\right)^2 + \left(\frac{\partial g_3}{\partial u_1}\right)^2 = \lambda^2(u_1, u_2),$$

$$\left(\frac{\partial g_1}{\partial u_2}\right)^2 + \left(\frac{\partial g_2}{\partial u_2}\right)^2 + \left(\frac{\partial g_3}{\partial u_2}\right)^2 = \lambda^2(u_1, u_2), \quad (41)$$

$$\frac{\partial g_1}{\partial u_1}\frac{\partial g_1}{\partial u_2} + \frac{\partial g_2}{\partial u_1}\frac{\partial g_2}{\partial u_2} + \frac{\partial g_3}{\partial u_1}\frac{\partial g_3}{\partial u_2} = 0.$$

Our partial solution can be worked into an actual solution by letting $g_3$ satisfy

$$\left(\frac{\partial g_3}{\partial u_1}\right)^2 = \lambda^2(u_1, u_2) - \left(\frac{\partial g_1}{\partial u_1}\right)^2 - \left(\frac{\partial g_2}{\partial u_1}\right)^2,$$
$$(42)$$

with all other derivatives of $g_3$ vanishing. Hence $g_3$ is constant in all directions except the $u_1$ direction so that, geometrically speaking, the $u_1$ direction is bent and warped by the embedding into the additional $x_3$ dimension.

To summarize, compared to conformal mappings, with dimension-changing conformal embeddings the number of equations in the system remains the same but the number of functions available to satisfy them increases. This allows conformal embeddings to be much more expressive than the fixed set of conformal mappings, but also prevents an explicit classification and parameterization of all conformal embeddings.

## C. Experimental Details

### C.1. Synthetic Spherical Distribution

**Model** The injective part of the model $\mathbf{g}$ was composed of a padding layer, SCT, orthogonal transformation, and translation (See App. B.2 for the definition of SCT). The bijective part $\mathbf{h}$ stacked three sets of Glow-style blocks, each having an ActNorm, Invertible $1 \times 1$ Convolution, and Affine Coupling layer.

**Training** We trained the reconstruction loss with a 100-epoch manifold-warmup phase for $\mathbf{g}$, then trained the mixed loss function in Eq. (6) with the end-to-end log-likelihood for 100 epochs, and finally 100 epochs with the manifold model fixed to fine-tune the density. We used a batch size of 100 and a learning rate of 0.001.

**Data** For illustrative purposes we generated a synthetic dataset from a known distribution on a spherical surface embedded in $\mathbb{R}^3$. The sphere is a natural manifold with which to demonstrate learning a conformal embedding with a CEF, since we can analytically find suitable maps $\mathbf{g} : \mathbb{R}^2 \to \mathbb{R}^3$ that embed the sphere[3] with Cartesian coordinates describing both spaces. For instance consider

$$\mathbf{g} = \left( \frac{2r^2 z_1}{z_1^2 + z_2^2 + r^2}, \frac{2r^2 z_2}{z_1^2 + z_2^2 + r^2}, r\frac{z_1^2 + z_2^2 - r^2}{z_1^2 + z_2^2 + r^2} \right),$$
$$(43)$$

where $r \in \mathbb{R}$ is a parameter. Geometrically, this embedding takes the domain manifold, viewed as the surface $x_3 = 0$ in $\mathbb{R}^3$, and bends it into a sphere of radius $r$ centered at the origin. Computing the Jacobian directly gives

$$\mathbf{J}_\mathbf{g}^T \mathbf{J}_\mathbf{g} = \frac{4r^4}{(z_1^2 + z_2^2 + r^2)^2}\mathbf{I}_2, \qquad (44)$$

which shows that $\mathbf{g}$ is a conformal embedding (Eq. (4)) with $\lambda(\mathbf{z}) = \frac{2r^2}{z_1^2 + z_2^2 + r^2}$. Of course, this $\mathbf{g}$ is also known as a *stereographic projection*, but here we view its codomain as all of $\mathbb{R}^3$, rather than the 2-sphere.

With this in mind it is not surprising that a CEF can learn an embedding of the sphere, but we would still like to study how a density confined to the sphere is learned. Starting with a multivariate Normal $\mathcal{N}(\boldsymbol{\mu}, \mathbf{I}_3)$ in three dimensions we drew samples and projected them radially onto the unit sphere. This mimics the much more complicated distribution given by integrating out the radial coordinate from the

---

[3]Technically the "north pole" of the sphere $(0, 0, 1)$ is not in the range of $\mathbf{g}$, which leaves a manifold $\mathbb{S}^2\backslash\{\text{north pole}\}$ that is topologically equivalent to $\mathbb{R}^2$.

standard Normal distribution

$$p_{\mathcal{M}}(\phi, \theta) = \int_0^\infty \frac{1}{(2\pi)^{3/2}} \exp\left\{ -\frac{1}{2} \left( r^2 - \right. \right.$$

$$\left. \left. 2r\left(\cos\phi\sin\theta,\ \sin\phi\sin\theta,\ \cos\theta\right) \cdot \boldsymbol{\mu} + \|\boldsymbol{\mu}\|^2 \right) \right\} r^2 dr. \tag{45}$$

With the shorthand $\mathbf{t} = (\cos\phi\sin\theta,\ \sin\phi\sin\theta,\ \cos\theta)$ for the angular direction vector, the integration can be performed

$$p_{\mathcal{M}}(\phi, \theta) = \frac{1}{2^{5/2}\pi^{3/2}} e^{-\|\boldsymbol{\mu}\|^2/2} \Big( 2\mathbf{t}\cdot\boldsymbol{\mu} +$$

$$\sqrt{2\pi}\left((\mathbf{t}\cdot\boldsymbol{\mu})^2 + 1\right) e^{(\mathbf{t}\cdot\boldsymbol{\mu})^2/2} \left( \mathrm{erf}\left(\mathbf{t}\cdot\boldsymbol{\mu}/\sqrt{2}\right) + 1 \right) \Big). \tag{46}$$

This distribution is visualized in Fig. 2 for the parameter $\boldsymbol{\mu} = (-1, -1, 0)$.

### C.2. CelebA

**Models** The baseline's embedding $\mathbf{g}$ is a Glow-style network of 3 levels and 2 steps per level: the output of each scale is reshaped into $8 \times 8$, and all scales are concatenated. We then apply an invertible $1 \times 1$ convolution, and project the input down to 1536 dimensions. Since this network is not conformal, joint training is intractable, so it must be trained sequentially.

On the other hand, both CEFs use the same conformal architecture for $\mathbf{g}$: a series of $2 \times 2$ and $1 \times 1$ Householder convolutions interspersed with Conditional Orthogonal convolutions, padding layers, shifts, and scales. The basic architecture follows. Between every layer, trainable scaling and shift operations were applied.

$\mathbf{x}\ (3 \times 64 \times 64) \to 4 \times 4$ Householder Conv
$\qquad \to 1 \times 1$ Householder Conv
$\qquad \to 1 \times 1$ Conditional Orthogonal Conv
$\qquad \to$ Pad Channels$(48, 24)$
$\qquad \to 2 \times 2$ Householder Conv
$\qquad \to 1 \times 1$ Householder Conv
$\qquad \to 1 \times 1$ Conditional Orthogonal Conv
$\qquad \to$ Pad Channels$(96, 48)$
$\qquad \to 2 \times 2$ Householder Conv
$\qquad \to 1 \times 1$ Householder Conv
$\qquad \to 1 \times 1$ Conditional Orthogonal Conv
$\qquad \to$ Pad Channels$(192, 96)$
$\qquad \to \mathbf{u}\ (96 \times 8 \times 8)$

**Training** S-MF and S-CEF were sequentially trained, meaning $\mathbf{g}$ was trained with a reconstruction loss in a 10-epoch

manifold-warmup phase, and then $\mathbf{h}$ was trained to maximize likelihood for 100-epochs. J-CEF was trained with a 10-epoch manifold warmup phase, and then the mixed loss function in Eq. (6) was optimized for 100 epochs with weights of 0.001 for the likelihood and 100 for the reconstruction loss.

All models were trained with the Adam optimizer (Kingma & Ba, 2015) with learning rate $1 \times 10^{-5}$ and batch size 32. Using a single NVIDIA TITAN V, the sequential baseline and sequential CEF ran for 34 hours, while the joint CEF ran for 65 hours.