# OpenReview forum: "Conformal Embedding Flows: Tractable Density Estimation on Learned Manifolds"
_ICML.cc/2021/Workshop/INNF — INNF+ 2021 spotlighttalk_

### Official Review · Reviewer_H3bH · 2021-06-11

**Rating:** Accept
**Confidence:** 3

**Summary:**

This work proposes a solution to the problem of normalizing flows with data that lives on an unknown low-dimensional manifold. Existing approaches share the drawback that exact densities cannot be calculated through the dimensionality reducing step. By constraining this reduction to the class of conformal embeddings, with several examples given in the paper, the authors present an approach that offers tractable density estimation and is straightforward to train.

**Justification For Rating:**

The paper addresses a relevant problem and proposes a plausible and effective solution. It is easy to follow and, without rigorously checking the mathematical derivations, I did not see any mistakes in the thought process. The experiments clearly show that the idea works in practice. I think conformal embeddings are an interesting contribution to the toolbox of normalizing flows and thus merit publication in the community.

---

### Official Review · Reviewer_qQD2 · 2021-06-14

**Rating:** Accept
**Confidence:** 4

**Summary:**

This work proposes a novel normalizing flow for modelling data supported on an unknown manifold. The proposed model includes conformal embedings which are a special case of orthogonal mapping allowing for easier manifold learning and for explicit computation of data log-probability. Although experiments show that the conformal embedding flow has limited expressiveness, this model looks promising for future research. The paper is written extremely well, the main idea and all important details are explained very clearly, experiments highlight main advantages and limitations of the model showing that proposed idea works and can be developed in the future. It was my pleasure to review this paper, thanks to the authors for a good work.

Minor comment: It would be good to add a sentence about the meaning of FID score in the section 4.2 (at least whether higher value is better or lower one). That would make easier to understand the comparison for a broader audience.

**Justification For Rating:**

The idea is interesting, the study is sound and presented extremely well. This work will be very interesting for community

---

### Decision · Program_Chairs · 2021-06-14

Accept (spotlight talk)